# Recurrent and Metastatic Head and Neck Cancer: Mechanisms of Treatment Failure, Treatment Paradigms, and New Horizons

**DOI:** 10.3390/cancers17010144

**Published:** 2025-01-05

**Authors:** William T. Barham, Marshall Patrick Stagg, Rula Mualla, Michael DiLeo, Sagar Kansara

**Affiliations:** 1Department of Otolaryngology-Head and Neck Surgery, Louisiana State University Health Sciences Center, New Orleans, LA 71103, USA; wbarh1@lsuhsc.edu (W.T.B.); rmuall@lsuhsc.edu (R.M.); mdileo@lsuhsc.edu (M.D.); 2Department of Oncology, Our Lady of the Lake Regional Medical Center, Baton Rouge, LA 70809, USA; marshall.stagg@fmolhs.org

**Keywords:** metastatic head and neck cancer, immunotherapy, squamous cell carcinoma

## Abstract

This review aims to summarize and discuss recent advances in the treatment of recurrent or metastatic head and neck squamous cell carcinoma. Our current understanding of the biology of metastasis and treatment failure, as well as historical and ongoing clinical trials aimed at treating recurrent or metastatic head and neck cancer, is reviewed. Current management concepts are discussed, and important insights gleaned from treatment failure are also explored. Finally, the impact of artificial intelligence on the treatment of recurrent or metastatic head and neck squamous cell carcinoma is assessed, and future treatment modalities integrating this emerging technology into existing treatment paradigms are considered.

## 1. Introduction

Head and neck squamous cell carcinoma, arising on the mucosal surfaces of the upper aerodigestive tract and traditionally associated with tobacco exposure, is among the most common cancers globally, with greater than 660,000 new cases and 325,000 deaths annually [1]. Recurrent or metastatic head and neck squamous cell carcinoma (R/M HNSCC), which fails initial treatment or has spread beyond its initial site of occurrence, contributes greatly to the morbidity and mortality associated with this disease. Advances in the detection and treatment of R/M HNSCC have improved treatment response rates, patients’ quality of life, and overall survivorship. Although the landscape of this disease has evolved with the advent of human papillomavirus positive (HPV+) oropharynx scc, our understanding of the disease continues to progress as we uncover mechanisms of treatment failure and metastasis. Historically, risk factors associated with decreased overall survival in HNSCC patients with metastatic disease include a lack of response to prior chemotherapy, prior radiation therapy, hypopharynx or oral cavity primary site, weight loss > 5%, and decreased cytologic differentiation [2].

Classical approaches to treating R/M HNSCC utilized chemotherapy with agents such as cisplatin, methotrexate, and bleomycin, with an eventual progression to combined therapies that exhibited higher response rates, though they increased toxicities [3,4,5,6]. With the advent of the epidermal growth factor receptor (EGFR) inhibitor cetuximab, platinum-based chemotherapy augmented with cetuximab further improved response rates, progression-free survival, and overall survival [7]. More recently, novel approaches utilizing the body’s own immune system via immunotherapeutic methods have revolutionized the treatment of R/M HNSCC. Agents such as programmed cell death protein 1 (PD-1), programmed death ligand 1 (PD-L1), and cytotoxic T-lymphocyte-associated protein 4 (CLTA-4) inhibitors, in combination or as monotherapy, are finding increasing use as first-line agents for patients with R/M HNSCC. A better understanding of the molecular changes occurring in normal cells spurring their conversion to metastatic carcinoma has propelled the development of newer immunotherapy-based treatment modalities, such as harnessing T cell exhaustion, genetically targeted therapies, and peptide- or protein-based vaccines. These emerging therapies also harness the body’s own immune system to eliminate neoplastic growth, thus avoiding the overt toxicity of chemotherapeutic, systemic, or radiation-based approaches. A better understanding of the basic biology of metastasis, treatment failure, and immune evasion and escape has evolved rapidly in recent years, though much is still to be learned. In this review, we hope to provide a broad overview of biological mechanisms underlying R/M HNSCC and its successful treatment, as well as a summary of published and ongoing trials investigating systemic therapy of R/M HNSCC with a particular emphasis on the burgeoning field of cancer immunotherapy.

## 2. Biology of Metastasis

Cancer, defined as the unregulated proliferation of abnormal host cells endowed with the potential to invade locally or spread to other parts of the body, may manifest in numerous ways [8,9]. Metastatic disease is often a later complication of cancerous growth, corresponding to disease that has spread to a secondary site beyond its initial origin. In order for normal tissue to progress to cancerous tissue, it undergoes dysplastic change and, in the case of metastatic cancer, loses the cellular structures anchoring it in place to surrounding healthy tissue. One of these intrinsic components of cellular adhesion is E-cadherin, which contributes specifically to cell–cell adhesion and is one of the classic cell surface markers lost in the cancer biogenesis model of the epithelial–mesenchymal transition (EMT) [10]. Though EMT plays an essential role in development by allowing primitive cells the freedom to organize into distinct cellular layers during gastrulation and in wound healing by facilitating cellular infiltration and change during the wound healing process, this activity may also inappropriately endow a cancer cell with the ability to invade other tissues. During progression to metastasis, cancer cells will also gain the ability to proliferate continuously without cell cycle checkpoint inhibition. During this process, increases and decreases in characteristic cellular biomarkers often reflect the new capabilities of these tumor cells. For example, FOS1 is a super-enhancer that increases the transcription of pro-growth related proteins and is classically increased during EMT, while increased expression of matrix metalloproteases (MMPs) heralds the ability of these neoplastic cells to degrade the extracellular matrix responsible for circumscribing them to their native location in the body [10]. Though the overall mutational etiology of HNSCC is broad as compared to other malignancies, such as prostate cancer or leukemia, whole exome sequence analyses exhibit near universal inactivation of *TP53*, heralding disruption of the normal pathway of squamous differentiation via numerous different initial insults [11].

The transcriptome, which refers to all coding and non-coding RNA transcripts generated by a cell, has been used to help researchers describe expression changes unique to cancer cells. Transcriptome analyses aimed at describing the mRNAs corresponding with cellular proliferation, response to hypoxia, and epithelial differentiation in cancer cells support the role of EMT in driving metastasis [12]. HNSCC with partial epithelial to mesenchymal transcription (p-EMT) patterns have been localized to the leading edge of primary tumors, where they likely drive the local and distal expansion of neoplastic growth. The degree of p-EMT may even independently predict nodal metastasis and tumor grade [12]. Other transcriptomic analyses have examined the role of epithelial growth factor receptor (EGFR) itself in EMT in advanced HNSCC, where mitogen activated protein kinase (MAPK) mediated proliferation and cell survival follows initial migration and invasion [13]. Variability in response to EGFR inhibition and expression of EGFR in HNSCC also suggests that EGFR-independent pathways, such as the m-TOR dependent Akt pathway, often concomitantly contribute to the local progression of HNSCC prior to and following its metastatic spread [14].

Transcriptome analyses of the tumor microenvironment immediately surrounding neoplastic cells have also contributed greatly to our understanding of the mechanisms of neoplastic growth and HNSCC metastasis. Mito et al. described unique immune signatures surrounding normal versus neoplastic tissue, associating lymphocyte rich local environments in HPV+ HNSCC with improved survival, versus myeloid and dendritic cell enriched local environments associated with shorter overall survival and HPV-negative HNSCC [15]. Mounting research attention on the tumor immune microenvironment (TIME) has led to an increased interest in better understanding the inflammatory milieu surrounding viral versus tobacco and alcohol carcinogen-driven HNSCC, which may undergo EMT and metastatic changes via slightly different immune escape pathways [16]. For example, tumor cell analysis of T-cell surface cell markers CD3, CD8, FOXP3, PD-1, PD-L1, and pancytokeratin by immunofluorescence, whole-exome sequencing, and RNA sequencing shows that active smoking decreases IFN-mediated signaling and cytotoxic T cell activity in immune-mediated tumor destruction [17]. Though more research is needed to fully understand the interplay between immune evasion and escape and the progression of local neoplasia, these findings suggest that harnessing and enhancing endogenous immune function would restore the body’s natural response to R/M HNSCC. Additionally, these findings allude to the potential importance of smoking cessation in patients being treated for R/M HNSCC.

Following neoplastic initiation, promotion, and conversion, lymphovascular invasion (LVI) and perineural invasion (PNI) also function as key drivers of metastasis by allowing neoplastic cells to invade along lymphatic or venous structures, as well as nerve sheaths. From a clinical perspective, in T3-4 oral squamous cell carcinoma, both PNI and LVI are independently and significantly associated with neck metastasis, distant metastasis, and decreased 5 year survival rate [18]. In HNSCC, the degree of lymph node metastasis has been described as an independent predictor of survival outcomes following secondary analyses of NRG/RTOG 9501, NRG/RTOG 0234, and EORTC 22931 [19]. Efforts to better understand genomic changes underlying LVI are underway. Study of HNSCC patients within the National Cancer Institute’s Cancer Genome Atlas has led to description of genes that code for small molecular agents that are upregulated in cells undergoing LVI, shedding light on potential mediators driving LVI and potential future targets for small molecular inhibitors in preventing the progression to LVI [20]. Karmakar and colleagues identified microRNAs miR-203a-3p, mir-10a-5p, and miR-194-5p as being associated with LVI in HNSCC using a random forest approach of The Cancer Genome Atlas’s HNSC database and hypothesized their involvement in fatty acid oxidation pathways and lymphangiogenic gene expression [21]. Schwann cell dysfunction may contribute to the progression of PNI [22]. Ultimately, a better understanding of the precise mechanisms driving LVI and PNI may allow us to identify actionable targets and opportunities for intervention.

## 3. Biology of Treatment Failure

Because HNSCC may exhibit resistance to all treatment modalities used to address it, including radiotherapy, chemotherapy, and surgery, as well as immunotherapy approaches, a significant body of literature has been dedicated to better understanding the molecular processes underlying treatment failure in R/M HNSCC patients. In head and neck cancer, radiotherapy resistance is thought to occur via numerous mechanisms, including cancer stem cells, increased DNA repair capability, enhanced reactive oxygen species scavenging, EMT, and abnormal programmed cell death [23]. The tumor microenvironment of HNSCC has also been found to be enriched in dysregulated/hyperactive fibroblasts, tumor associated macrophages, and myeloid-derived suppressor cells (MDSCs), which are thought to be partially responsible for the tissue proliferation and suppressed endogenous immune response in resistant scc [23].

In addition to identification of the mediators of radiotherapy resistance at the whole cell level, new research has implicated altered expression of intracellular mediators as playing a role in impaired cancer cell ubiquination in radiated scc [24]. Increased expression of peroxidases, such as peroxiredoxin 6, which are responsible for reactive oxygen species (ROS) scavenging, reduce radiation induced ROS and prevent cell apoptosis in scc [25]. Radiation induced chemokine secretion, specifically CCL20, has also been implicated in the body’s anti-tumor immunity mechanisms via its influence on the behavior of regulatory T cells (Tregs), which are attracted to radiated tissue by CCL20 and play a role in the suppression of anti-tumor immunity, and thus radiotherapy resistance [26]. Cancer stem cells, which can renew and regenerate all types of tumor tissue in the face of radiotherapy, are also thought to play a large role in radioresistance, because stem cells themselves are inherently radioresistant. Efforts to conduct radiobiological assays may allow for identification of targetable surface markers or intracellular mediators upregulated in these cells relative to healthy tissues [27,28].

Resistance to systemic therapy with the chemotherapeutic agents cisplatin, 5 fluorouracil, or paclitaxel, in combination with RT, or in combination with the EGFR-targeted therapeutic cetuximab, also contributes to treatment failure in advanced HNSCC. Resistance to cisplatin, the most commonly used chemotherapeutic agent, may occur via intrinsic or adaptive mechanisms that modify cell susceptibility to intracellular drug levels, DNA susceptibility to damage, and anti-inflammatory response to the generation of reactive oxygen species. Some specific examples of these resistance mechanisms include modification of cisplatin uptake via copper membrane transporters 1 and 2, increased expression rates of DNA adduct repair via nucleotide excision repair associated proteins (ERCCs), and modulation of cell detoxification responsive transcription factor nuclear factor erythroid 2-related factor 2 (NF2) [29]. Significant heterogeneity in cancer stem cells may also drive systemic therapy resistance, and the high mutational burden of HPV-negative HNSCC, specifically, may drive cisplatin therapy failure in a clonal survival, tumor relapse, and subsequent regeneration sequence of events not dissimilar from radiotherapy treatment failure [29,30]. Among influential cell types in the TIME, head and neck cancer-associated fibroblasts (HNCAFs) have been one cell subtype identified as particularly relevant to driving therapy response, serving as just one example of a potential biomarker of treatment resistance within the inflammatory milieu [31]. Epigenetic reprogramming, altered chromatin states, and transient increases in IGF-1 receptor signaling in response to tumor cell stressors have also been hypothesized as drivers of transcriptional plasticity in treatment-resistant cell populations [32,33].

## 4. History of Systemic Therapy and Progression to Immunotherapy

From the first demonstration of the efficacy of cisplatin in HNSCC in 1977, great progress has been made in the use of systemic agents in the treatment of HNSCC refractory to or incompatible with surgical excision or radiotherapy [3]. The first trial examining cisplatin, bleomycin, and placebo showed a response rate of 25% and increased 10-week mean overall survival times in the cisplatin group, and from that point forward, trials have rarely included a placebo group [3]. In 1985, a comparison of cisplatin with methotrexate in patients with recurrence of HNSCC following surgery or radiotherapy helped to establish the non-superiority of any one chemotherapeutic agent in subsequent clinical trials, though the addition of infusional 5-FU to cisplatin treatment regimens further improved observed treatment response rates in 1994, ushering in doublet therapy [4,5,6]. Subsequent studies investigating the taxanes (paclitaxel, docetaxel) in the 1990s showed similar efficacy in doublet therapy with cisplatin, though they remained non-superior with respect to median survival times in palliative treatment. Triplet therapies TIP (paclitaxel, ifosfamide, cisplatin) and TIC (paclitaxel, ifosfamide, carboplatin) remain as a less studied and more cytotoxic alternative option, with no improvement in mean overall survival time relative to doublet therapy [34,35,36]. However, triplet therapy with paclitaxel, cisplatin, and 5-FU has been used with success in the context of induction therapy [37]. In 2008, following the advent of the EGFR-targeted biologic cetuximab and its proof of concept as a non-inferior monotherapy in 2007, the EXTREME trial showed that the addition of cetuximab to a platinum doublet significantly improves response rates (36% vs. 20%), progression-free survival (5.6 mo versus 3.3 mo), and overall survival in recurrent HNSCC (10.1 mo versus 7.4 mo) [7,38].

Following the description of the immunoglobulin PD-1’s involvement in programmed cell death in 1992 and the subsequent use of cancer-specific monoclonal antibody rituximab for CD20 B cell non-Hodgkin lymphoma in 1997, immunotherapies have carved out a niche among cancer treatment protocols [39,40,41]. In 2010, a phase III trial examining the treatment of metastatic melanoma with cytotoxic T-lymphocyte-associated antigen 4 (CLTA-4) inhibitor versus glycoprotein 100 (gp100) protein peptide vaccine found an increased overall survival (median overall survival of 10.1 mo versus 6.4) in the CLTA-4 groups relative to the gp100-only controls [42]. Since this initial proof-of-concept study in patients with unresectable, metastatic cancer, the use of immunomodulators for this population has been a burgeoning field of research interest. Indeed, the overall 5-year survival rates for metastatic melanoma, formerly near zero, now approach 50%.

In essence, immunomodulatory agents regulate T-cell activation and their response to cancer-associated antigens distinguishing these neoplastic cells from normal tissues, following initial immune recognition [43,44]. In the midst of costimulatory immune pathways mediating programmed cell death in cells recognized as non-self, the programmed cell death protein 1 (PD-1) system promotes self-tolerance, which functions to guard the body against autoimmunity. Simply put, the PD-1 and programmed death ligand 1 (PD-L1) system functions as a break on the immune system’s natural immunosurveillance and destruction of non-self cells (Figure 1). While PD-L1 is expressed preferentially on antigen-presenting cells and tumor cells and appears to increase with the degree of metastasis, PD-1 is expressed on monocytes, natural killer cells, macrophages, and T cells. Inhibiting the crosstalk between PD-1/PD-L1 thus allows the body to resume its natural T cell-mediated anti-tumor inflammatory response in the context of tumor cells gaining peripheral tolerance inappropriately. While CTLA-4 is hypothesized to inhibit T cell proliferation early in the immune response, at the level of the lymph nodes, PD-1 is thought to suppress T cells later in the immune response, at the level of peripheral tissues [45].

A major landmark in the field of immunotherapy for cancer occurred in 2016, when the phase III twin trials Checkmate 017 and Checkmate 057 investigated the use of the fully humanized IgG4 anti-PD-1 monoclonal antibody in patients with progressive non-small cell lung cancer [46,47]. The results of these twin trials on patients treated with PD-1 inhibitors following platinum-containing systemic therapy saw the first FDA approval of an immunotherapeutic agent to target neoplasia. Pembrolizumab, granted breakthrough therapy designation, was recommended at a dose of 200 mg intravenously every three weeks, with the noted common side effects of hypothyroidism, anorexia, nausea, constipation, fatigue, dyspnea, and cough, with more rare side effects including pneumonitis, colitis, hepatitis, and nephritis. More rarely, encephalopathy and Guillain-Barre syndrome may be observed. Efforts to further understand the differences underlying variable treatment response to immune checkpoint blockade (ICB) across many cancer types has advanced the theory that high mutational burden and high immune signature expression may be associated with an improved ICB response, perhaps via increased cancer-associated antigens, while numerous copy number alterations may be associated with a decreased ICB response [48].

Updated evidence-based guidelines addressing immunotherapy and biomarker testing have aimed to streamline these agents’ use into clinical practice. In 2021, the National Comprehensive Cancer Network (NCCN) and, in 2022, the American Society of Clinical Oncology (ASCO) released a series of recommendation statements and guidelines addressing PD-L1 immunohistochemistry testing, use of combined positive score (CPS), tumor mutational burden (TMB) testing, and PD-1 inhibitor monotherapy and combination therapy in platinum-naïve and refractory patients [41,49]. Per the NCCN, pembrolizumab/platinum/5-FU is emphasized as the preferable first line option for R/M HNSCC with no surgical or radiotherapeutic option following the results of the KEYNOTE-048 trial [50]. In patients with a higher CPS (>1), pembrolizumab monotherapy may be recommended. Conversely, lower CPSs (<1) necessitate a combination therapy with additional platinum and 5-FU. In patients with platinum-refractory R/M HNSCC, CPS status is not used and PD-1 inhibitors pembrolizumab and nivolumab may be offered first-line to all patients with recurrent, unresectable, or metastatic disease without a surgical or radiotherapeutic alternative, regardless of CPS status. This recommendation follows the results of the KEYNOTE-012 and KEYNOTE-040 trials in platinum-refractory HNSCC patients [51,52]. Among other systemic therapy options for non-nasopharyngeal H&N cancers, numerous single-agent and combination therapies may be used, depending on the individual patient’s goals of therapy. Additionally, per the ASCO, TMB testing may be pursued in R/M HNSCC cancer patients without a CPS or in patients with rare tumors, and a TMB score of >10 is interpreted as being consistent with a clinical benefit of PD-1/PD-L1 system inhibition [2]. However, TMB’s use as a biomarker for immune checkpoint blockade efficacy appears to have no correlation with improved treatment outcomes [53]. Finally, radiation therapy was recommended as safe with concurrent immunotherapy in patients requiring palliation or for local control [49].

## 5. Newly Published and Ongoing Trials: PD-L1 Inhibitors, PD-1 Inhibitors, and CTLA-4 Inhibitors

In 2016’s KEYNOTE-012 phase Ib trial, the safety, tolerability, and anti-tumor activity of pembrolizumab were first established in patients with R/M HN scc regardless of PD-L1 expression status, with 17% of patients having grade 3 or 4 adverse events (primarily, transaminitis, hyponatremia, and one case of drug-related rash) [51]. In 2019’s landmark KEYNOTE-040 trial, pembrolizumab was compared to methotrexate, docetaxel, or cetuximab for R/M HNSCC in a randomized, open-label, phase 3 study in patients who had disease progression during or after platinum-containing treatment for recurrent or metastatic disease (or both), or whose disease recurred or progressed within 3–6 months of previous multimodal therapy containing platinum for locally advanced disease, in a proof-of-concept study supporting pembrolizumab as first line therapy for patients with platinum-refractory disease [52] (Table 1).

In 2019, the randomized, open-label KEYNOTE-048 phase III trial of patients with untreated locally incurable R/M HNSCC examined three treatment arms, pembrolizumab alone, pembrolizumab plus platinum and 5-fluorouracil (pembrolizumab with chemotherapy), or cetuximab plus platinum and 5-fluorouracil (cetuximab with chemotherapy), and based on the findings of this study, pembrolizumab plus platinum and 5-FU shortly thereafter joined the ranks of first-line therapies for R/M HNSCC [50]. Patients in this trial were stratified by their PD-L1 expression, p16 status, and performance status. PD-L1 expression combined positive score (CPS) influenced overall survival, supporting the importance of immunotherapy and immune targeting in the treatment of R/M HNSCC, and emphasized the importance of the PD-1/PD-L1 pathway in mediating immune evasion in HNSCC. In 2020, KEYNOTE-158 further investigated the role of tumor mutational burden (TMB) in patient selection for immunotherapy, with significantly different objective response rates for the high TMB (≥10 mutations per megabase) [30%] versus the low TMB (<10 mutations per megabase) [13%] subgroups, informing the inclusion of TMB testing in ASCO guidelines in patients without a CPS or in rare tumors [55].

The survival benefit of immunotherapy relative to other systemic options appears durable. This is supported by the 4-year follow-up of the KEYNOTE-048 trial, where first-line pembrolizumab and pembrolizumab–chemotherapy exhibited continued survival benefit relative to cetuximab–chemotherapy in R/M HNSCC patients [56]. Interestingly, in comparing a PD-1 inhibitor relative to PD-L1 inhibitors in R/M HNSCC, a decreased risk of death with PD-1 inhibitor relative to PD-L1 inhibitor has been observed, though anti-PD-L1 therapy may be associated with better outcomes in females and HPV-positive or locally recurrent disease [57]. In 2020’s EAGLE trial, the PD-L1 inhibitor durvalumab (n = 240), durvalumab plus tremelimumab (a CTLA-4 inhibitor) (n = 247), and the standard of care, defined by the authors as cetuximab, a taxane, methotrexate, or a fluoropyrimidine (n = 249), were compared in patients with R/M HNSCC. With respect to the primary outcome of overall survival, no difference was observed, though addition of CLTA-4 inhibitor did increase 12–24 month survival rates and response rates [58].

One phase II trial examining a combination therapy for R/M HNSCC using EGFR inhibition as well as PD-1 inhibition has also shown encouraging results, with a 45% overall response rate at 6 months with a relatively favorable side effect profile. These findings help to support the use of PD-1/PD-L1 therapy concurrent with other targeted systemic therapies [54]. Unfortunately, additional investigation into combination nivolumab (another PD-1 inhibitor) with stereotactic body radiotherapy in R/M HNSCC has found no evidence of objective response rate improvement, which remains relatively low for PD-1 inhibitors in general in R/M HNSCC [59].

### Insights from Locally Advanced HNSCC

In the literature addressing locally advanced HPV-negative HNSCC, attempts at reducing tumor growth with the addition of stereotactic body radiation therapy to single-dose durvalumab have added to the initially disappointing results of immunotherapy as an adjuvant in locally advanced disease [60,61]. However, in patients disqualified from high-dose cisplatin therapy, the phase II trial GORTEC 2015-01 PembroRad suggested non-inferiority of pembrolizumab with concurrent radiotherapy relative to cetuximab with concurrent radiotherapy in locally advanced (LA) HNSCC [62]. The relative inefficacy of immunotherapy in the concurrent setting has been corroborated in several studies. Mechanistically, this may be in part due to the fact that radiotherapy (and surgery) disrupts the lymphatic structures that are essential for immune response, thus attenuating immunotherapy effect when administered in a concurrent setting. These findings have paved the way for immunotherapy to be used in the neoadjuvant setting in several clinical trials [63].

## 6. Insights from Immunotherapy and Predictive Biomarkers

Efforts to harness the host immune system response in the fight against metastatic cancer have led to great advances in our understanding of the function of tumor-infiltrating lymphocytes and the function of immune checkpoint inhibitors. Unfortunately, a majority of patients do not benefit from immunotherapy treatment at the onset (primary resistance) or relapse after initially responding (acquired resistance) [64]. The mechanism of treatment failure via primary resistance is thought to occur both via tumor cell intrinsic factors, which refer to intracellular changes within the neoplastic cells themselves, and extrinsic changes, which refer to components other than tumor cells in the tumor immune microenvironment (TIME), such as T-regs, MDSCs, and M2 macrophages. Primary resistance via tumor intrinsic intracellular mechanisms include numerous alterations, such as changes in mitogen-activated protein kinase (MAPK) pathway, WNT/β-catenin signaling, IFN-gamma signaling, and loss of T-cell antigen presentation [65]. In patients with advanced lung cancer, next-generation sequencing (NGS) has determined STK11 and KEAP1 mutations to be relatively frequent drivers of treatment non-response to PD-1 inhibition despite a high TMB; in the future, NGS methods may be helpful in uncovering similar predictors of PD-1 susceptibility in R/M HNSCC [66]. Furthermore, G protein-coupled receptors (GPCRs), which have been closely studied physiologically and are now understood to contribute to tumor intrinsic oncogenic signaling, also appear to play an important role in primary resistance and may constitute a potential target to circumvent primary resistance in patients receiving immunotherapy [67]. Across solid tumor types in numerous non-H&N cancer KEYNOTE trials, IFN-gamma release and its effect on PD-L1/PD-1 expression also predictably play a role in pembrolizumab responder status. Interestingly, some PD-1 inhibitor non-responders appear to fail to respond via immunosuppressive pathways outside of the PD-1 signaling axis [68]. Elevated plasma interleukin-8 (IL-8) may also inhibit antigen presentation in tumor cells and appears to decrease response to checkpoint inhibition therapy in patients undergoing atezolizumab treatment in metastatic renal cell cancer [69].

Research aimed at better understanding host-related factors in treatment failure has highlighted the role that tumor infiltrating lymphocytes (TILs) may play in treatment failure, though their role in driving metastasis, particularly to lymph nodes, is not clear. The inflammatory milieu, comprised of TILs with their associated cytokines, should ideally also be considered when targeting PD-1, as inhibition of PD-1 would most efficiently be accomplished with synergistic inhibition of those same chemokines that induce PD-L1 localization to the cell surface. For example, the inhibition of myeloid-derived suppressor cells (MDSCs) from interacting with tumor cells may increase the efficacy of immune checkpoint blockade by preventing MDSCs (and associated chemokines) from participating in tumor cell invasion, angiogenesis, and further metastasis [70,71]. Additionally, in patients with metastatic melanoma being treated with the CLTA-4 inhibitor ipilimumab, peripheral blood mononuclear cells (PBMCs) appear to be increased in the serum of patients with severe metastatic disease relative to regional metastases, supporting their role as a potential prognostic biomarker for immunotherapy resistance [72]. TIME-associated macrophages are yet another extrinsic mediator of immunotherapy resistance. PD-L1 expression on tumor associated macrophages increases with disease progression and is thought to allow for immune tolerance by inhibiting T cell activation in non-H&N cancer [73]. Another biomarker category, extracellular vesicles, represents yet another potentially groundbreaking prognostic biomarker whose levels in the host’s body fluid (saliva) have been associated with the prognosis of patients with oral cavity squamous cell carcinoma [74].

Though a staple of immunotherapy relative to other treatment modalities is the longevity of the response to treatment, it is becoming clear that with an increased length of therapy and use of these agents for a plethora of malignant conditions, including R/M HNSCC, more relapsing patients will be identified. KEYNOTE-048 has already noted progressive disease following immune therapy for R/M HNSCC; investigation into tumor response in patients with advanced melanoma suggests acquired resistance rates may range from 18 to 26% in previously immunotherapy-naïve patients [56,75]. Some of the mechanisms that have been proposed to mediate acquired resistance in immunotherapy patients include tumor-intrinsic mutational changes, altered T cell antigen presentation, or loss of T cell function following long-term antigen-mediated immune activation, loosely termed T cell exhaustion [64,76]. Work by Zaretsky et al. highlighted JAK-1/JAK-2 mutations as particularly advantageous for PD-1 inhibitor acquired immune evasion via the inhibition of IFN-gamma mediated signaling and antigen presentation via MHC-1 surface expression in patients with metastatic melanoma [77]. It may be that acquired resistance to ACT and the overall T cell response may be mediated by a switch to Th-2 helper-type immune response in the long term [78].

Insights from investigations into mechanisms of immunotherapy treatment failure have led to the advent of tumor mutational burden (TMB) and microsatellite instability-high/deficient mismatch repair (MSI-H/dMMR) as putative biomarkers of PD-1 checkpoint inhibitor response in HPV-negative HNSCC, in addition to the already well-established role of PD-1 expression as a biomarker for PD-1 inhibitor response [79]. The development of immunohistochemical assays to tailor immunotherapy treatment to other predictive biomarkers, such as PD-L1 expression levels, are also being pursued in the treatment of other cancer types, such as non-small cell lung cancer [80]. With respect to prognosticating immune-related adverse events during R/M HNSCC immunotherapy, PD-L1 polymorphisms rs4143815 and rs2282055 have been advanced as potential predictive biomarkers [81].

Traditional carcinogen-derived HNSCC and HPV-related HNSCC both result in epigenetic changes that can lead to abnormal cellular biology [82,83]. These aberrations occur in the absence of DNA alterations and can include non-coding RNAs, DNA methylation, and histone modifications [83]. Changes at this level can predict response to treatment. For example, hypermethylation of death-associated protein kinase (DAPK) and associated transcriptional silencing can predict resistance to cetuximab [84]. In addition, cisplatin resistance may be mediated by hypermethylation of several genes including *CRIP1*, *GOS2*, *MLH1*, and *S100*; treatment with the demethylating agent decitabine restored cisplatin sensitivity in HNSCC cell lines [85]. Further, methylation of the PD-1 promoter results in a marked decrease in infiltrating CD8+ T-lymphocytes, thus attenuating natural immunity to carcinogenesis [86]. Starzer and colleagues found via microarray analysis that DNA methylation signatures were bioinformatically and statistically correlated with response to checkpoint inhibitor therapy, confirming methylation signature as a predictive biomarker [87].

Ultimately, understanding a patient’s specific TIME via biomarkers will allow for a precision approach with immunotherapy in combination with targeted agents, thus maximizing response and improving survival outcomes. In the future, standardization of predictive biomarkers in the TIME and uniform application of these biomarkers to tailored therapy in clinical trials may allow for further progress in managing R/M HNSCC.

## 7. New Horizons: Artificial Intelligence in Precision Oncology

Understanding of the mechanisms of H&N cancer recurrence and metastasis continues to progress rapidly. However, the process of assessing patient factors and the TIME and using that to inform treatment decisions and predict treatment response lags far behind. Artificial intelligence (AI) has gained much attention recently as a potentially useful tool in precision oncology. Although in its nascency, AI has evolved rapidly, serving as a revolutionary opportunity to move H&N cancer care forward.

Clinically, AI can help predict early treatment failure. Photographs taken during routine surveillance visits can be examined by deep learning systems to detect early recurrences. Indeed, one AI model was able to reliably identify suspicious oral lesions using high-resolution photographic images [88]. Pathologically, AI can predict malignant or premalignant lesions by analyzing TILs and tumor cytologic parameters, including nuclear size, shape, and chromatin distribution [89].

When clinical, pathological, and radiological models are synthesized, AI can very accurately predict poor outcomes. Recently, Cai and colleagues formulated an AI multilayer perceptron model that could accurately predict postoperative recurrence of oral cancer [90]. Another study implemented deep learning to devise a multimodal convolutional neural network combining clinical features and CT and PET findings in order to identify H&N cancer patients at high risk of locoregional recurrence [91]. Further, a recent systematic review showed that the application of an AI-based radiomics model offered useful quantitative information that could be used to predict recurrence, metastasis, and oncologic survival [92]. This may be transformative in how clinicians counsel patients regarding treatment and prognosis.

AI can also identify physiologic or genomic biomarkers to detect recurrent H&N malignancy. AI can process exponentially large sets of biologic and physiologic data, thus identifying variations in correlative proteomic or metabolic information [93,94]. At the genomic level, AI can analyze gene expression, epigenetic variability, and tumor mutational burden to identify unique genetic biomarkers in HNSCC [95,96]. This same analysis can be performed in metabolomics and proteomics, resulting in a rich array of protein biomarkers that can serve as potential drug targets for novel drug discovery [97].

Once a patient with recurrent or metastatic HNSCC has been identified, AI can assist with developing an individualized treatment regimen. AI algorithms can simultaneously integrate patient history, tumor characteristics, and genetic and proteomic information to predict a regimen that might be the most effective and least toxic [98,99]. Further, monitoring of physiologic markers, radiologic tumor size, and adverse effects can allow for a refinement in treatment plan, providing alternatives if progression is identified [100].

Ultimately, AI can serve as a novel tool to 1. detect early recurrence, 2. provide a patient-specific risk assessment to inform counseling, 3. provide novel drug targets and personalize treatment, and 4. predict the treatment response in a patient-specific manner. When this process is iterated for each new patient with R/M HNSCC using AI, treatment regimens will become exponentially more efficacious and less toxic (see Figure 2).

## 8. Conclusions

R/M HNSCC remains a challenge for both patients and the health care providers treating them. Advances in our understanding of mechanisms of treatment failure and disease progression have opened the door for breakthroughs in immunotherapy. Further, vital information is gleaned from non-responders. Next-generation sequencing technology, as well as the increasing number of fully humanized monoclonal antibodies in development to target specific neoplastic mediators, represents a powerful combination of tools at the disposal of researchers and clinicians aiming to decrease metastatic tumor burden, stabilize metastatic disease, or prevent recurrence. Ultimately, a holistic understanding of tumor and patient factors (demographic, genomic, proteomic, and otherwise) combined with AI precision oncology approaches will allow for a targeted and unique approach to each patient. In this setting, revolutions in treatment will be inevitable.

## Figures and Tables

**Figure 1 cancers-17-00144-f001:**
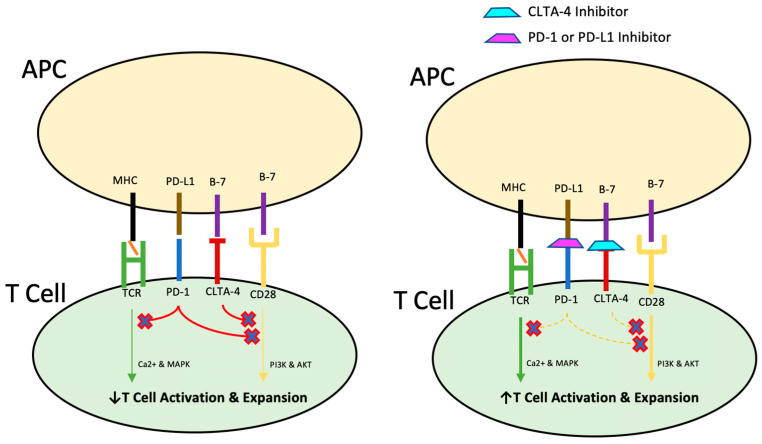
T cell activation and expansion occurs in response to costimulatory signaling from the Major Histocompatibility Complex (MHC)’s presentation of foreign epitopes on the antigen-presenting cell (APC) to the T Cell Receptor (TCR) in the context of positive costimulation by B7. Programmed death ligand 1 (PD-1) and cytotoxic T-lymphocyte-associated protein 4 (CLTA-4) attenuate this activation to prevent host autoimmunity to native cells. Thus, PD-1 or CTLA-4 inhibitors prevent T cell attenuation, which augments the immune response to abnormal cells.

**Figure 2 cancers-17-00144-f002:**
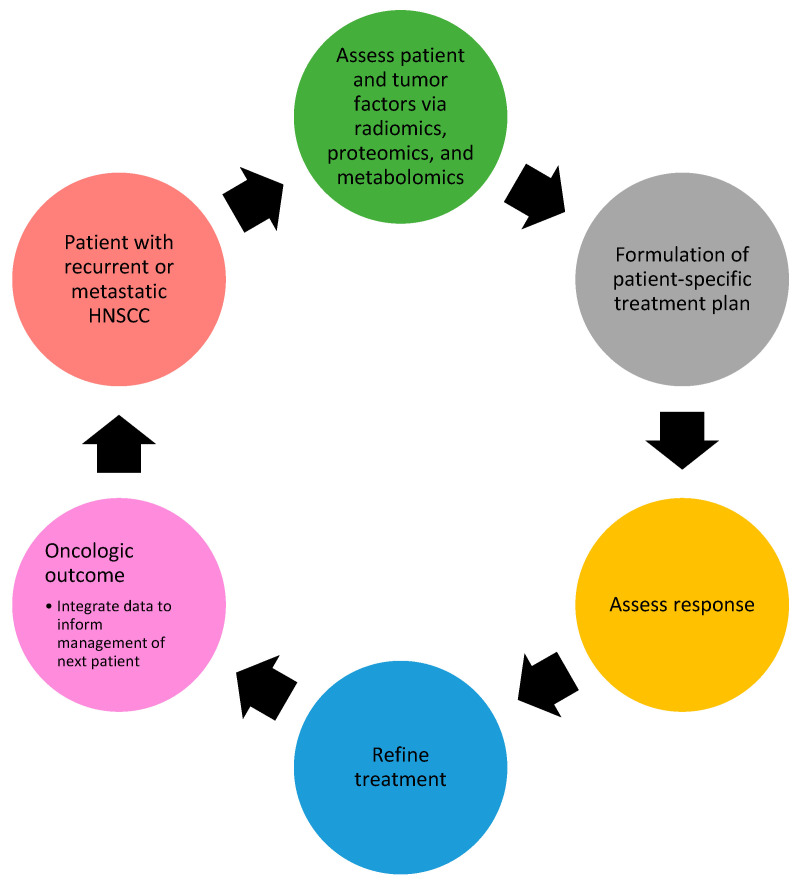
Stepwise approach to refining treatment of HNSCC using artificial intelligence to evolve management.

**Table 1 cancers-17-00144-t001:** Recently published and ongoing trials examining immunotherapy in the treatment of locally advanced or recurrent/metastatic head and neck squamous cell carcinoma.

Trial Name	Study Comparison	Years Active; Phase of Study	# of Patients	Outcome Measures	Summary
KEYNOTE-040	Pembrolizumab versus methotrexate, docetaxel, or cetuximab	Phase 3 2014–2016	247	Median overall survival for pembrolizumab was 8.4 months versus 6.9 months with standard of care	A clinically significant increase in overall survival was observed in the pembrolizumab monotherapy treatment group relative to standard of care treatment group.
KEYNOTE-048	Pembrolizumab versus pembrolizumab–chemotherapy versus cetuximab–chemotherapy	Phase 3 2015–2017	882	Pembrolizumab with chemotherapy improved overall survival relative to cetuximab plus chemotherapy (13.0 versus 10.7 months); pembrolizumab alone also improved overall survival relative to cetuximab plus chemotherapy in population with CPS of 20 or more (14.9 months versus 10.7 months)	First-line pembrolizumab and pembrolizumab–chemotherapy continued to demonstrate survival benefit versus cetuximab–chemotherapy in recurrent/metastatic head and neck squamous cell carcinoma; patients responded well to subsequent treatment after pembrolizumab-based therapy
GORTEC 2015-01 PembroRad	Pembrolizumab versus cetuximab concurrent with radiotherapy in patients with locally advanced squamous cell carcinoma of head and neck unfit for cisplatin	Phase 2 2016–2017	133	Locoregional recurrence was 59% with cetuximab plus once-daily radiotherapy versus 60% with pembrolizumab plus once-daily radiotherapy	Pembrolizumab concomitant with RT did not improve tumor control relative to cetuximab plus RT in patients with R/M HNSCC
Sacco et al., 2021 [54]	Pembrolizumab plus cetuximab for R/M HNSCC	Phase 2 2017–2019 Additional cohorts remain open	33	45% overall response rate (95% CI = 28–92) in platinum-resistant or platinum-ineligible patients or patients with R/M HNSCC with no prior immunotherapy or EGFR inhibition	PD-1 inhibitor pembrolizumab plus EGFR inhibitor cetuximab showed 45% response rate for R/M HNSCC
KEYNOTE-055	Pembrolizumab for patients refractory to cetuximab or platinum for R/M HNSCC; no comparison group	Phase 2 2014–2017	171	16% overall response rate (95% CI = 11–23%) in platinum- and cetuximab-refractory patients, defined as disease progression within 6 months of platinum and cetuximab treatment	In patients with refractory R/M HNSCC, overall response and survival rates were improved relative to treatment with single agent, cetuximab, afatinib, or methotrexate; there was no improvement in progression-free survival
CHECKMATE 141	Nivolumab (PD-1 inhibitor) versus standard single-agent systemic therapy (methotrexate, docetaxel, or cetuximab) for patients with R/M HNSCC refractory to platinum	Phase 3 2014–2016	361	Median overall survival in PD-1 inhibition group was 7.5 months versus 5.1 months for standard therapy in patients with disease progression within 6 months of platinum therapy	In patients with platinum-refractory R/M HNSCC, patients who took PD-1 blockade therapy had a significantly improved median overall survival on the scale of approximately 2 months
EAGLE	Durvalumab versus durvalumab plus tremelimumab versus standard of care (cetuximab, a taxane, methotrexate, fluoropyrimidine)	Phase 3 2015–2020	763	Median overall survival of 9.8 months (95% CI: 4.3–14.1) for durvalumab, 4.8 months for dual therapy including tremelimumab, and 9.0 months for standard of care	No survival benefit for single-agent (PD-1 inhibition) immunotherapy or double-agent (PD-1 and CTLA-4 inhibition) immunotherapy over triple-agent standard of care with cetuximab, selected taxanes, and fluoropyrimidine
IMCISION	Neoadjuvant nivolumab versus nivolumab plus ipilimumab in candidates for resectable HNSCC	Phase Ib/II	32	Neoadjuvant immune checkpoint blockade achieved complete or near-complete pathological response as measured by sequencing testing on paired tumor biopsies	The resolution of hypoxia may play a role in the success of an immune checkpoint blockade in HNSCC as an outcome biomarker
KESTREL	Durvalumab +/− tremelimumab versus standard of care	Phase III 2015–2017	823	Durvalumab and durvalumab with tremelimumab were not superior to EXTREME for overall survival in patients with high PD-1 expression; however, combination therapy may prolong duration of response	Immunotherapy may have utility in increasing the long-term benefit of the response in existing standard-of-care R/M HNSCC treatment modalities
HAWK	Durvalumab monotherapy	Phase III 2016–2017	112	Adverse effects, quality of life metrics, progression-free survival, median overall survival	Treatment with single therapy using anti-PD-1 or PD-L1 monoclonal antibodies led to overall survival improvement in patients with R/M HNSCC
LEAP-10	Pembrolizumab and lenvatinib	Phase III Discontinued	500 (estimated)	Objective response rate, progression-free survival, duration of response, safety, tolerability	Incorporation of scheduled tumor imaging assessment following treatment into management paradigm
INDUCE-3	GSK3359609 (feladilimab) versus pembrolizumab	Phase II/III Ongoing	314	Overall survival, progression-free survival, safety, and tolerability	No utility of adding feladilimab to treatment regimen
CHECKMATE 714	Nivolumab plus ipilimumab versus nivolumab	Phase II 2016–2019	425	Median duration of response, objective response rate, adverse events	Nivolumab and ipilimumab were generally well tolerated and provided patients with a theoretic improvement in the treatment of R/M HNSCC
CHECKMATE 651	Nivolumab plus ipilimumab versus standard of care (EXTREME = cetuximab, cisplatin, 5-FU)	Phase III 2017–2020	947	Overall survival, duration of clinical treatment	No statistically significant improvement in the OS versus the EXTREME trial

## Data Availability

Not applicable.

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
