# Peer review of "Recurrent and Metastatic Head and Neck Cancer: Mechanisms of Treatment Failure, Treatment Paradigms, and New Horizons"

_cancers, 2025, doi:10.3390/cancers17010144_

Round 1

Reviewer 1 Report

Comments and Suggestions for Authors

The authors have thoroughly  reviewed the current understanding of HNSCC resistance to chemoradiotherapy, targeted therapy, and immunotherapy. Although only few topics are picked up from basic science, overall quality is acceptable.

1. Fig. 1. Another B7 bar should be added above CTLA-4.

2. Line 401. MDSC cannot be detected in serum instead of PBMC.

Author Response

The authors have thoroughly  reviewed the current understanding of HNSCC resistance to chemoradiotherapy, targeted therapy, and immunotherapy. Although only few topics are picked up from basic science, overall quality is acceptable.

  1. Fig. 1. Another B7 bar should be added above CTLA-4. 
          1. Response: Thank you for the suggestion. This has been added. 

2. Line 401. MDSC cannot be detected in serum instead of PBMC.

2. Response: Thank you. Line 414 has been edited to reflect the suggestion.

Reviewer 2 Report

Comments and Suggestions for Authors

The manuscript entitled “Recurrent and Metastatic Head and Neck Cancer: Mechanisms of Treatment Failure, Treatment Paradigms, and New Horizons”, authored by William Barham et al., presents a review of the of biological mechanisms of treatment failure and metastasis in HNSCC and a summarized list of ongoing clinical trials in the management of the disease. The authors also present new perspectives in artificial intelligence (AI) assistance in HNSCC patients management.

Minor changes: 

It would be desirable to use the usual long-established acronym HNSCC instead of H&N scc.

In line 392 TILs should be used instead of TIL.

Mayor changes:

The authors start the analysis of HNSCC metastasis and recurrence biology by addresing EMT. Authors may include at the beginning a brief description of mutations associated with HNSCC dissemination and other genetic phenomena related to this.

Line 133 contains references to cutaneous SCC, which usually is not included in HNSCC analysis, since it has a very different pathogenesis. Please reevaluate and search for HNSCC references in vascular invasion.

In line 229 PD-1/PD-L1 and CTL4 are mentioned in the context of immunomodulatory agents that enhance immune cell recognition. This is not quite so. These checkpoint inhibitors act after immune recognition by modulating T cell activation and response.

Authors present TILs in the context of immunotherapy treatment failure. Since their involvement in metastasis (as predictors), particularly lymph node metastases, is less clear, it would be important to address this issue.

Also, automated or AI assisted analysis of TILs in histological images should be addressed as the possible use of mutation prediction based on histological images analysis.

Finally, Extracellular vesicles (EVs) hold significant potential in the analysis and treatment of HNSCC as biomarkers or in relation to immunosuppression and should be mentioned in the manuscript.

Author Response

The manuscript entitled “Recurrent and Metastatic Head and Neck Cancer: Mechanisms of Treatment Failure, Treatment Paradigms, and New Horizons”, authored by William Barham et al., presents a review of the of biological mechanisms of treatment failure and metastasis in HNSCC and a summarized list of ongoing clinical trials in the management of the disease. The authors also present new perspectives in artificial intelligence (AI) assistance in HNSCC patients management.

Minor changes: 

It would be desirable to use the usual long-established acronym HNSCC instead of H&N scc.

In line 392 TILs should be used instead of TIL.

Thank you for the suggestions. We have edited the text to reflect suggestions.

Major changes:

The authors start the analysis of HNSCC metastasis and recurrence biology by addresing EMT. Authors may include at the beginning a brief description of mutations associated with HNSCC dissemination and other genetic phenomena related to this.

Thank you for the suggestion. Lines 78-117 address these concerns. 

Line 133 contains references to cutaneous SCC, which usually is not included in HNSCC analysis, since it has a very different pathogenesis. Please reevaluate and search for HNSCC references in vascular invasion.

Thank you for the suggestion. Lines 145-153 are edited to address this concern. 

In line 229 PD-1/PD-L1 and CTL4 are mentioned in the context of immunomodulatory agents that enhance immune cell recognition. This is not quite so. These checkpoint inhibitors act after immune recognition by modulating T cell activation and response.

Thank you. Lines 240-242 are edited to address this concern. 

Authors present TILs in the context of immunotherapy treatment failure. Since their involvement in metastasis (as predictors), particularly lymph node metastases, is less clear, it would be important to address this issue.

Thank you. Lines 404-405 address this concern. 

Also, automated or AI assisted analysis of TILs in histological images should be addressed as the possible use of mutation prediction based on histological images analysis.

Thank you. Lines 480-482 are edited to address this concern.

Finally, Extracellular vesicles (EVs) hold significant potential in the analysis and treatment of HNSCC as biomarkers or in relation to immunosuppression and should be mentioned in the manuscript.

Thank you. Lines 420-422 address this concern.

Reviewer 3 Report

Comments and Suggestions for Authors

The authors conduct an updated narrative review on the understanding of the biological mechanisms involved in treatment failure and metastasis, also addressing published and ongoing clinical trials in managing metastatic head and neck cancer. The review is clear, well-structured, and provides a comprehensive topic overview. However, despite being a highly relevant field, the theme is not entirely new, and other reviews have already addressed similar issues. I believe the review could benefit from a section discussing the most common cancer subtypes in this region. It would be interesting to explore how the authors, while reviewing the literature, perceive how the scenario might change in the coming years and decades, considering new advances in treatment and the understanding of biological mechanisms.

Author Response

The authors conduct an updated narrative review on the understanding of the biological mechanisms involved in treatment failure and metastasis, also addressing published and ongoing clinical trials in managing metastatic head and neck cancer. The review is clear, well-structured, and provides a comprehensive topic overview. However, despite being a highly relevant field, the theme is not entirely new, and other reviews have already addressed similar issues. I believe the review could benefit from a section discussing the most common cancer subtypes in this region. It would be interesting to explore how the authors, while reviewing the literature, perceive how the scenario might change in the coming years and decades, considering new advances in treatment and the understanding of biological mechanisms.

Thank you for the suggestions. Squamous cell carcinoma is the most common type of metastatic H&N cancer. We feel that there may be several changes in the coming years and decades - specifically the text addresses neoadjuvant therapy in lines 373, and further in the Precision Oncology section lines 471-510. We hope that ultimately, assessment of each patient's unique radiomics, proteomics, and metabolomics, results in a unique management strategy. Response is assessed and this data is integrated each time via an iterative process by which treatment becomes incredibly refined.